# Nearly one in every six HIV-infected children lost from ART follow-up at Debre Markos Referral Hospital, Northwest Ethiopia: A 14-year retrospective follow-up study

Yitbarek Tenaw Hibstie[1], Getiye Dejenu Kibret[2,3], Asmare Talie[2], Belisty Temesgen[1], Mamaru Wubale Melkamu[1], Animut Alebel[2,3]*

1 Debre Markos Referral Hospital, Debre Markos, Ethiopia, 2 College of Health Science, Debre Markos University, Debre Markos, Ethiopia, 3 School of Public Health, Faculty of Health, University of Technology Sydney, Ultimo, NSW, Australia

* animut.a23@gmail.com

**Data Availability Statement:** All relevant data are within the manuscript and its Supporting Information files.

## Abstract

### Background

Although antiretroviral therapy (ART) significantly improves the survival status and quality of life among human immunodeficiency virus (HIV)-infected children, loss to follow-up (LTFU) from HIV-care profoundly affecting the treatment outcomes of this vulnerable population. For better interventions, up-to-date information concerning LTFU among HIV-infected children on ART is vital. However, only a few studies have been conducted in Ethiopia to address this concern. Thus, this study aims to identify the predictors of LTFU among HIV-infected children receiving ART at Debre Markos Referral Hospital.

### Methods

An institution-based retrospective follow-up study was done among 408 HIV-infected children receiving ART at Debre Markos Referral Hospital between 2005 and March 15, 2019. Data were abstracted from the medical records of HIV-infected children using a standardized data abstracted checklist. We used Epi-Data Version 3.1 for data entry and Stata Version 14 for statistical analysis. The Kaplan-Meier survival curve was used to estimate the survival time. A generalized log-rank test was used to compare the survival curves of different categorical variables. Finally, both bi-variable and multivariable Cox proportional hazard regression models were used to identify the predictors of LTFU.

### Results

Of 408 HIV-infected children included in the final analysis, 70 (17.1%) children were LTFU at the end of the study. The overall incidence rate of LTFU among HIV-infected children was found to be 4.5 (95%CI: 3.5–5.7) per 100-child years of observation. HIV-infected children living in rural areas (AHR: 3.2, 95%CI: 2.0–5.3), having fair or poor ART drug adherence (AHR: 2.3, 95%CI: 1.4–3.7), children started ART through test and treat approach (AHR:

**Funding:** The author(s) received no specific funding for this work.

**Competing interests:** The authors have declared that no competing interests exist.

**Abbreviations:** AHR, Adjusted Hazard Ratio; AIDS, Acquired Immune Deficiency Syndrome; ART, Antiretroviral Therapy; CPT, Cotrimoxazole Preventive Therapy; HAART, Highly Active, Antiretroviral Therapy; HAZ, Height for Age Z-score; Hgb, Hemoglobin; HIV, Human Immunodeficiency Virus; IPT, Isoniazid Preventive Therapy; LTFU, Loss to Follow-up; NNRTI, Non-nucleoside Reverse Transcriptase Inhibitors; SSA, Sub-Saharan Africa; WAZ, Weight for Age Z-score; WHO, World Health Organization; WHZ, Weight for Height Z score.

2.7, 95%CI: 1.4–5.5), and children started protease inhibiter (PI)-based ART regimens (AHR: 2.2, 95%CI: 1.1–4.4) were at higher risk of LTFU.

## Conclusion

This study found that one in every six HIV-infected children lost form ART follow-up. HIV-infected children living in rural areas, having fair or poor ART drug adherence, started ART based on test and treat approach, and taking PI-based ART regimens were at higher risk of LTFU.

## Introduction

Though the concept of "child at risk" highly varies across the continents, in Africa, it commonly refers to socially disadvantaged children, including human immunodeficiency virus (HIV) exposed infants [1]. HIV is a global challenge for humankind's survival, as Sub-Saharan Africa (SSA) is profoundly affected. Globally, an estimated 1.7 million children (age<15 years) were living with HIV in 2018 [2]. In 2014, 87% of new HIV infections and 86% of AIDS-related deaths among children were from SSA [3, 4]. Ethiopia is one of the SSA countries with a high HIV prevalence; nearly 62,000 children were living with HIV in 2017 [5, 6]. Antiretroviral therapy (ART) improves the survival status and quality of life among HIV-infected children through viral load suppression and increasing CD4 counts [7]. ART dug must be taken continuously and on a daily base to be effective [8]. Otherwise, patients could develop treatment failure and end-up with death in the early ART phase [5].

A systematic review found that about 5–29% of the children living HIV under the age of 10 lost from ART or died within the first one year of ART initiation worldwide in 2011 [9]. Another study from Asia and Africa reported that loss to follow up (LTFU) among HIV-infected children was reported as low of 4.1% in Asia and as high of 21.8% in West Africa [10]. In SSA, the proportion of LTFU among HIV-infected children after two years of ART initiation ranged from 9% in Southern Africa to 21.8% in West Africa [10]. As per report of previous studies, LTFU among HIV-infected children in Ethiopia was ranged from 5.9% [11] to 15% [12].

The Sustainable Development Goals (SDGs) are targeted to end the epidemic of HIV/AIDS by 2030 [13, 14]. Retention in ART care is a key strategy to achieve this ambitious goal [15]. In this regard, different interventions have been implemented nationally and internationally. For example, a short message service (SMS) to remind HIV-patients who missed their appointments has been implemented in Malawi, Kenya, South Africa, Mozambique, Zimbabwe, Rwanda, and Zambia [16]. The World Health Organization (WHO) also recommends that age-appropriate disclosure and caregivers' regular support are essential to improve retention in ART care [5]. The orphans and vulnerable children (OVC) program in Africa, in collaboration with health care workers, is working persistently to trace individuals lost from HIV-care [17]. The Ethiopian government has also implemented adherence support through phone calls to trace patients lost from ART care [18].

The most commonly reported contributing factors for LTFU among HIV-infected children are: lack of caregivers' contact information, fear of stigma, forgetfulness, scheduling conflict, lack of access to transportation, privacy concerns, not disclosing HIV-status, weak follow-up at the ART clinic, advanced WHO clinical disease stage, malnutrition, and younger age [19, 20]. Similarly, the common risk factors increasing LTFU among HIV-infected Ethiopian

children are far from health institutions, lack of access to transportation, fear of stigma, not disclosing HIV-status, bedridden status, and lack of understanding as ART is a lifelong medication [21, 22].

Though discontinuation from ART care among children in Ethiopia is a serious public health concern, to our knowledge, only a few studies have been conducted to explore this problem. Therefore, this study aims to assess LTFU among HIV-infected children receiving ART at Debre Markos Referral Hospital. The results of this study will help policymakers and program planners working in the area of HIV/AIDS to design appropriate interventions in reducing LTFU in this vulnerable population. This study will also serve as an input for further prospective observational or interventional studies.

## Methods

### Study design, period, and setting

An institution-based retrospective follow-up study was conducted at Debre Markos Referral Hospital between 2005 and March 15, 2019. Debre Markos town is found 300 km from Addis Ababa, the capital city of Ethiopia. The town has one referral hospital and three public health centers. The hospital serves for more than 3.5 million people in East Gojjam Zone and neighboring Zones. It has been providing ART care and follow-up services since 2005. Currently, the ART clinic has two medical doctors, six nurses, three data clerks, one porter, one cleaner, five case managers, four CDC contract employees, and six adherence supporters. The hospital uses standardized monitoring and follow-up forms, adopted from the Ethiopian ART guideline. A total of 466 HIV-infected children on ART have been recorded since 2005. Of these, 326 children have an active monthly ART follow-up.

### Study participants

The records of all HIV-infected children, whoever started ART at Debre Markos Referral Hospital, were the source population. The records of all HIV-infected children receiving ART between 2005 to March 15, 2019, and whose charts were available during the data collection period were our study population. All HIV-infected children who had at least one month of ART follow-up between 2005 and March 15, 2019 were included. Whereas, children who had incomplete baseline records (i.e., CD4 count, WHO stages, CPT, IPT, and hemoglobin level), unknown outcomes, and transferred in from other health institutions without baseline information were excluded.

### Sample size and sampling procedures

This study included all HIV-infected children's records ever started ART between 2005 and March 15, 2019. First, the files of all HIV-infected children ever started ART between 2005 and March 15, 2019 were sorted by data collectors. Second, children who met the above-mentioned inclusion criteria were isolated. Third, after excluding all incomplete records, a total of 408 HIV infected children records were included in the final analysis. Lastly, data across 14 years were collected from the charts of each enrolled child.

### Data collection procedure and quality control

The data extraction checklist was adapted from a standardized ART intake and follow-up forms currently used in the ART clinic of Debre Markos Referral Hospital. The most recent clinical and laboratory tests at ART initiation were considered as baseline information. However, if pre-treatment laboratory tests were not recorded at ART initiation, laboratory tests

done within the first month of ART initiation were considered as baseline information. In the case of two laboratory tests done in the first month of ART, the most recent laboratory value was taken as baseline. Before data collection, the consistency between the data extraction checklist and the recording system was cheeked by taking some randomly selected charts, and necessary amendments were made. Three nurses currently working in the ART clinic of Debre Markos Referral Hospital were recruited as data collectors. Moreover, the data extraction checklist was carefully prepared from a standardized ART intake and follow-up forms to maintain data quality. Furthermore, training for data collectors and supervisor about the objectives, significances, and variables of the study was given. Lastly, the supervisor and principal investigators carefully monitored the completeness and consistency of the whole data collection process.

## Variables of the study

The dependent variable was the time to LTFU among HIV infected children after ART initiation. The independent variables were socio-demographic variables, clinical characteristics and laboratory tests, ART and other medications-related variables, and nutritional variables. **Socio-demographic variables** were age of the child, sex of the child, caregivers' age, caregivers' residence, marital status of the caregiver, family size, educational status of the caregiver, caregivers' relation for the child, and religion. **Clinical characteristics and laboratory tests** were WHO clinical staging, CD4 count/percentage, Hgb level, viral load, functional or developmental status, and baseline OIs. **ART and other medications-related variables** included baseline ART regimens, duration of ART, ART side effects, regimen change, treatment failure, taking IPT, taking CPT, and adherence to ART. Nutritional status included weight for age (W/age), height for age (H/age), and weight for height (W/height).

## Definition of variables

**LTFU** was recorded when HIV infected children missed their appointments from one month to three months [5].

**Censored** was considered when HIV-infected children had died, formally transferred to other health institutions, or still alive and on ART at the end of the study.

**ART Adherence** was classified as good, fair, and poor. Good was recorded when the child took $\geq$ 95% or missed $\leq$ three pills of monthly dose. Fair was recoded when the child took 85–94% or missed four to eight pills of monthly dose. Finally, poor was recorded when the child took <85% or missed $\geq$ nine pills of monthly dose.

Children who had CD4 cell counts < 1500/mm$^3$ or 25% for age < 12 months, CD4 cell counts < 750/mm$^3$ or < 20% for age 12–35 months, CD4 cell counts < 350/mm$^3$ or < 15% for age 36–59 months, and CD4 cell counts < 200/mm$^3$ or < 15% for age $\geq$ 60 months were classified as CD4 counts or percentage (%) below the threshold [23].

**Child developmental status** was classified as appropriate (able to attain milestones for age), delayed (failure to attain milestones for age); and regressed (loss of what has been attained for age). According to the recent Ethiopian ART guideline, developmental status has the following components: language, psychosocial, fine and gross motor skills, and cognition [5].

Weight for Age (W/Age) Z-score < -2 SD, Height for Age (H/Age) Z-score < -2 SD and Weight for height (W/H) Z-score < -2 SD were considered to represent moderate underweight, stunting, and wasting respectively [24, 25].

W/Age Z-score < -3 SD, H/Age Z-score < -3 SD, and W/H Z-score < -3 SD were considered to represent severe underweight, stunting, and wasting respectively [24, 25].

## Data processing and analysis

We used EPI-Data Version 3.1for data entry and Stata Version 14 for analysis. The children's Z-scores (WAZ, HAZ, and WHZ/BAZ) were generated using WHO Anthro-Plus Version 1.04 and ENA smart software. Descriptive statistics for categorical variables were visualized using tables and graphs. Summary results of continuous variables were described using the measure of central tendency (mean or median) and dispersion (standard deviation or interquartile range). The Kaplan-Meier survival plot was used to estimate the survival time after ART initiation. A Generalized log-rank test was used to compare the survival curves between categorical variables. The necessary assumption of Cox-proportional hazard model was assessed using Schoenfeld residual test and log-log plot. We used the cox-Snell residuals test to check model fitness. Variables with p-values ≤ 0.25 in the bi-variable analysis were fitted into the multivariate analysis [26]. In the final model, variables with p-values < 0.05 were considered as statistically significant predictors. We used the adjusted hazard ratios (AHR) with their 95% confidence intervals and p-values to measure the strength of association and identify statistically significant predictors.

## Ethics considerations

Ethical letter was obtained from the Ethical Review Committee of College of Health Science, Debre Markos University. A permission letter was also written from the hospital general manager to HIV-care clinic focal person. As this was a retrospective study, informed consent from research participants was not feasible. Since the research was done by reviewing medical records, the individual patients were not subjected to any harm as far as confidentiality is kept. All collected data were coded and locked in a separate room, and computer data were secured by a personal password to maintain privacy. Lastly, names and unique ART numbers were not included in the data collection format, and the data were not disclosed to anyone other than the principal investigators.

# Results

## Socio-demographic characteristics participants

Among 466 retrieved HIV-infected children records, 58 (12.4%) were excluded due to incompleteness. Then, 408 HIV-infected children's records were considered for the final analysis. More than half (54.7%) were males, and nearly two-thirds (67.7%) were from urban areas. The mean age of participants at ART initiation was 6.9 years (SD: ±3.7 years), and the mean age of caregivers was 33.5 years (SD: ±9.5 years). Moreover, more than half (53.7%) of the parents were alive. Furthermore, about 66.6% of the children disclosed their HIV status, and the majority (85.5%) of children were from the family of orthodox religious followers (**Table 1**).

## Clinical, immunological, and nutritional characteristics of the children

More than half (52.7%) of the children had opportunistic infections (OIs) at ART initiation. About 52% of the study participants were classified as mild disease stage (WHO stage I and II). Nearly two-thirds (67.6%) of the children had CD4 counts or percentage above the threshold, and only 11.5% of the study children were anemic (Hgb< 10mg/dl). The majority (90.7%) of study participants started on NVP or EFV based ART drugs. More than half (60.8%) of the participants had ever taken past OI prophylaxis. Moreover, more than three-quarters (78.8%) of the children had good ART drug adherence in the last three months of ART follow-up. Furthermore, about 49.3%, 51.7%, and 75.5% of the children were underweight, stunted, and wasted, respectively (**Table 2**).

**Table 1. Sociodemographic characteristics of HIV-infected children receiving ART at Debre-Markos Referral Hospital, Northwest Ethiopia, 2019.**

| Variables | Frequency (N) | Percentage (%) |
|---|---|---|
| Sex | | |
| Male | 223 | 54.2 |
| Female | 185 | 54.3 |
| Age of the child (in months) | | |
| 0–35 month | 63 | 15.5 |
| 36–96 month | 207 | 50.7 |
| ≥ 97 month | 138 | 33.8 |
| Residence | | |
| Urban | 276 | 67.7 |
| Rural | 132 | 32.7 |
| Parents status | | |
| Both mother and father alive | 219 | 53.7 |
| One or both died | 189 | 46.3 |
| Age of the caregiver | | |
| 18–24 year | 51 | 12.5 |
| 25–34 year | 198 | 48.5 |
| 35–44 year | 113 | 27.7 |
| ≥ 45 year | 46 | 11.3 |
| Marital status of the caregiver | | |
| Married | 173 | 42.4 |
| Widowed/divorced | 136 | 33.3 |
| Others | 99 | 24.3 |
| Educational status of the caregiver | | |
| Unable to read and write | 148 | 36.3 |
| Able to read and write | 260 | 63.7 |
| Occupation of the caregiver | | |
| Farmer | 79 | 19.4 |
| Governmental | 86 | 21.1 |
| Non-governmental | 49 | 12.0 |
| House wife | 36 | 8.8 |
| Daily laborer | 89 | 21.8 |
| Merchant | 69 | 16.9 |
| Child HIV status disclosure (180) | | |
| Yes | 120 | 66.6 |
| No | 60 | 33.4 |
| Religion of the caregiver | | |
| Orthodox | 349 | 85.5 |
| Others | 59 | 14.5 |
| Relation of caregiver for the child | | |
| Parent | 348 | 85.3 |
| Other guardians | 60 | 14.7 |

## Incidence of loss to follow-up

In this study, the participants were followed for a minimum of two months and a maximum of 136 months. At the end of follow-up, 17.1% (95% CI: 13.7, 21.1%) of the children were LTFU from ART care. The total follow-up time of the entire cohort was 18,755 child-months of

**Table 2. Clinical, immunological, and nutritional characteristics of HIV infected children receiving ART at Debre-Markos Referral Hospital, Northwest Ethiopia, 2019.**

| Variables | Frequency (N) | Percentage (%) |
|---|---|---|
| Baseline OIs | | |
| Yes | 215 | 52.7 |
| No | 193 | 47.3 |
| Functional status (age ≥ 5 years)(287) | | |
| Working | 153 | 53.3 |
| Ambulatory | 113 | 39.4 |
| Bedridden | 21 | 7.3 |
| Developmental status (age < 5 year) (121) | | |
| Appropriate | 73 | 60.3 |
| Delayed | 41 | 33.9 |
| Regressed | 7 | 5.8 |
| WHO clinical staging | | |
| Stage I and II | 212 | 52.0 |
| Stage III and IV | 196 | 48.0 |
| CD4 counts or CD4% at baseline | | |
| Below the threshold | 132 | 32.4 |
| Above the threshold | 276 | 67.6 |
| Level hemoglobin | | |
| Anemic (< 10 mg/dl) | 47 | 11.5 |
| Non-anemic (≥ 10 mg/dl) | 361 | 88.5 |
| OI prophylaxis given | | |
| Yes | 248 | 60.8 |
| No | 160 | 39.2 |
| Baseline ART regimens | | |
| EFV or NVP based | 370 | 90.7 |
| PI based | 38 | 9.3 |
| OIs during follow-up | | |
| Yes | 120 | 29.4 |
| No | 288 | 70.6 |
| ART eligibility criteria | | |
| Immunologic or Clinical | 341 | 83.6 |
| Test and treat | 67 | 16.4 |
| ART adherence in last three months | | |
| Good | 321 | 78.7 |
| Fair/poor | 87 | 21.3 |
| ART drug side-effects | | |
| Yes | 20 | 4.9 |
| No | 388 | 95.1 |
| Regimen change | | |
| Yes | 85 | 20.8 |
| No | 323 | 79.2 |
| Treatment failure | | |
| Yes | 20 | 4.9 |
| No | 288 | 95.1 |
| Underweight | | |
| Normal | 207 | 50.7 |
| Underweight | 210 | 49.3 |
| Stunting | | |

(*Continued*)

**Table 2.** (Continued)

| Variables | Frequency (N) | Percentage (%) |
|---|---|---|
| Normal | 197 | 48.3 |
| Stunted | 211 | 51.7 |
| Wasting | | |
| Normal | 308 | 75.5 |
| Wasted | 100 | 24.5 |

observation. The incidence density of LTFU among HIV-infected children in this study was 4.5 (95% CI: 3.5, 5.7) per 100 child-years of observation. The incidence rate of LTFU within the first year of ART initiation was 8.2 (95% CI: 5.7, 11.7) per 100 child-years of observation, whereas the incidence rate of LTFU after one year of ART initiation was 3.3 (95% CI: 2.6, 4.6) months per 100 child-years of observation. The mean survival time of the entire follow-up was 108.8 months (95% CI: 103.1, 114.5 months) (**Fig 1**).

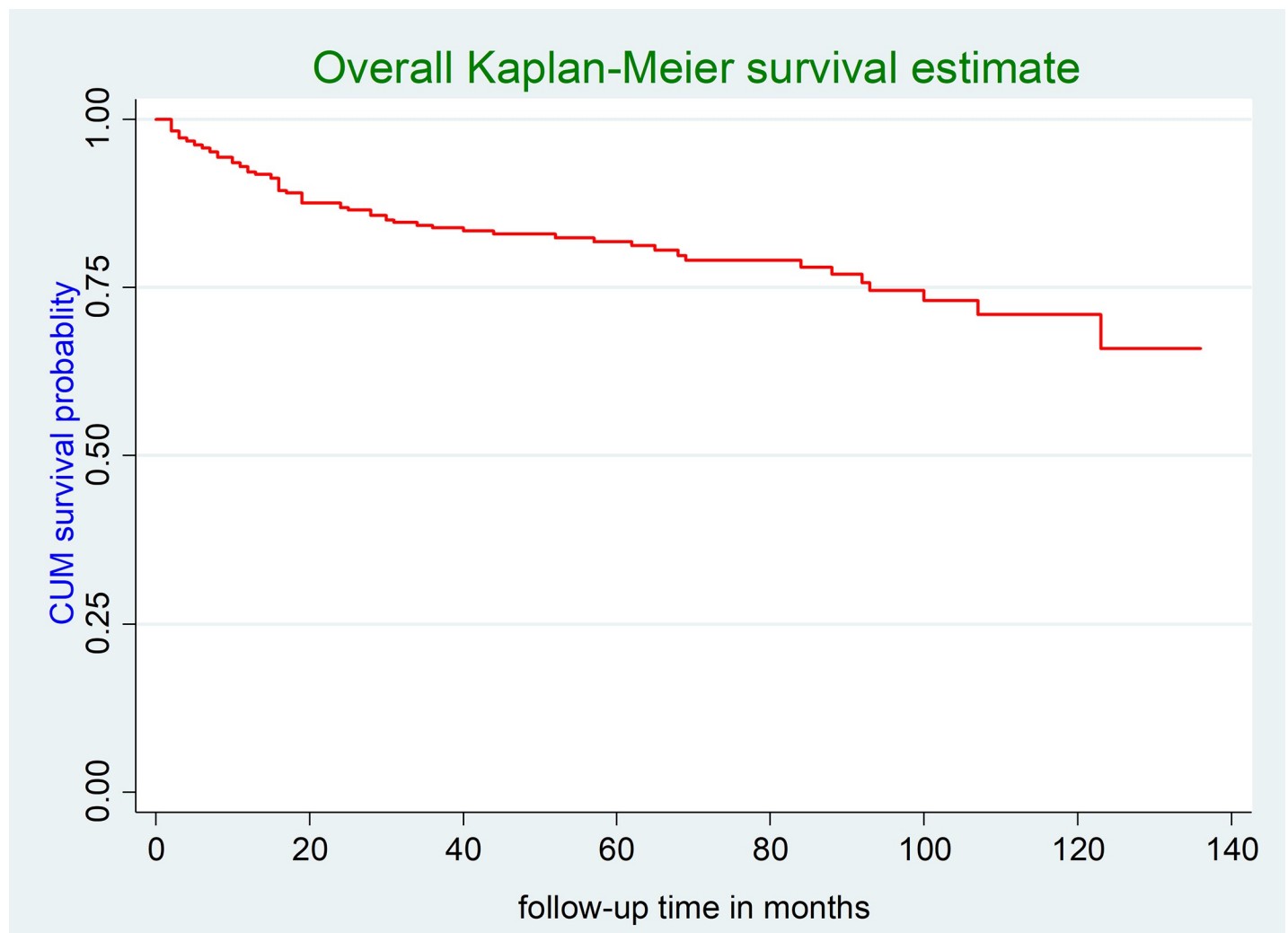

**Fig 1. The overall Kaplan-Meier survival curve of LTFU among HIV-infected children receiving ART at Debre-Markos Referral Hospital, Northwest Ethiopia, 2019.**

### Kaplan-Meier survival curves for different categorical predictors

The survival function of residence and ART drug adherence was compared using a generalized log rank test. Accordingly, the mean survival time of HIV-infected children receiving ART from urban areas was 119 months (95% CI: 113.1, 124.9 months); however, the mean survival time of HIV-infected children receiving ART from rural areas was 84.4 months (95% CI: 74.0–94.8 months). This difference was statistically significant (p-value < 0.001) (**Fig 2**).

The mean survival time of HIV infected children receiving ART who had good ART drug adherence was 115.1 months (95% CI: 109.0, 121.2 months); however, the mean survival time of HIV infected children receiving ART who had fair or poor ART drug adherence was 87.7 months (95% CI: 75.3, 99.9 months). This difference was statistically significant (p-value < 0.001) (**Fig 3**).

### Predictors of loss to follow-up

Ten variables (p-value < 0.25) were selected for the multivariable cox-regression analysis. In the multivariable analysis, only four variables were found to be statistically significant predictors of LTFU. Accordingly, the hazard of LTFU among HIV-infected children from rural areas

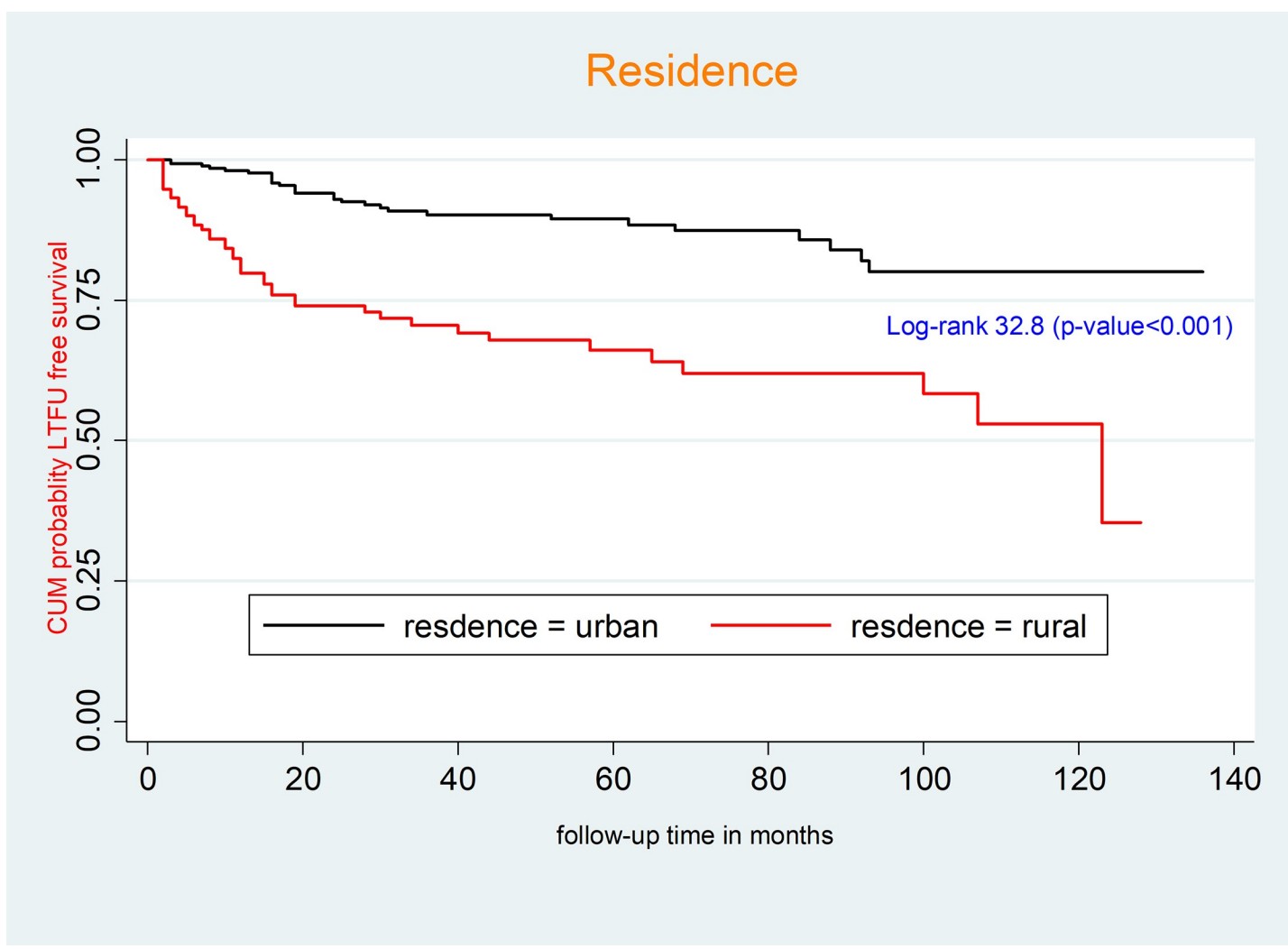

**Fig 2. Kaplan-Meier survival curves to compare LTFU among HIV-infected children receiving ART between rural and urban areas at Debre-Markos Referral Hospital, Northwest Ethiopia, 2019.**

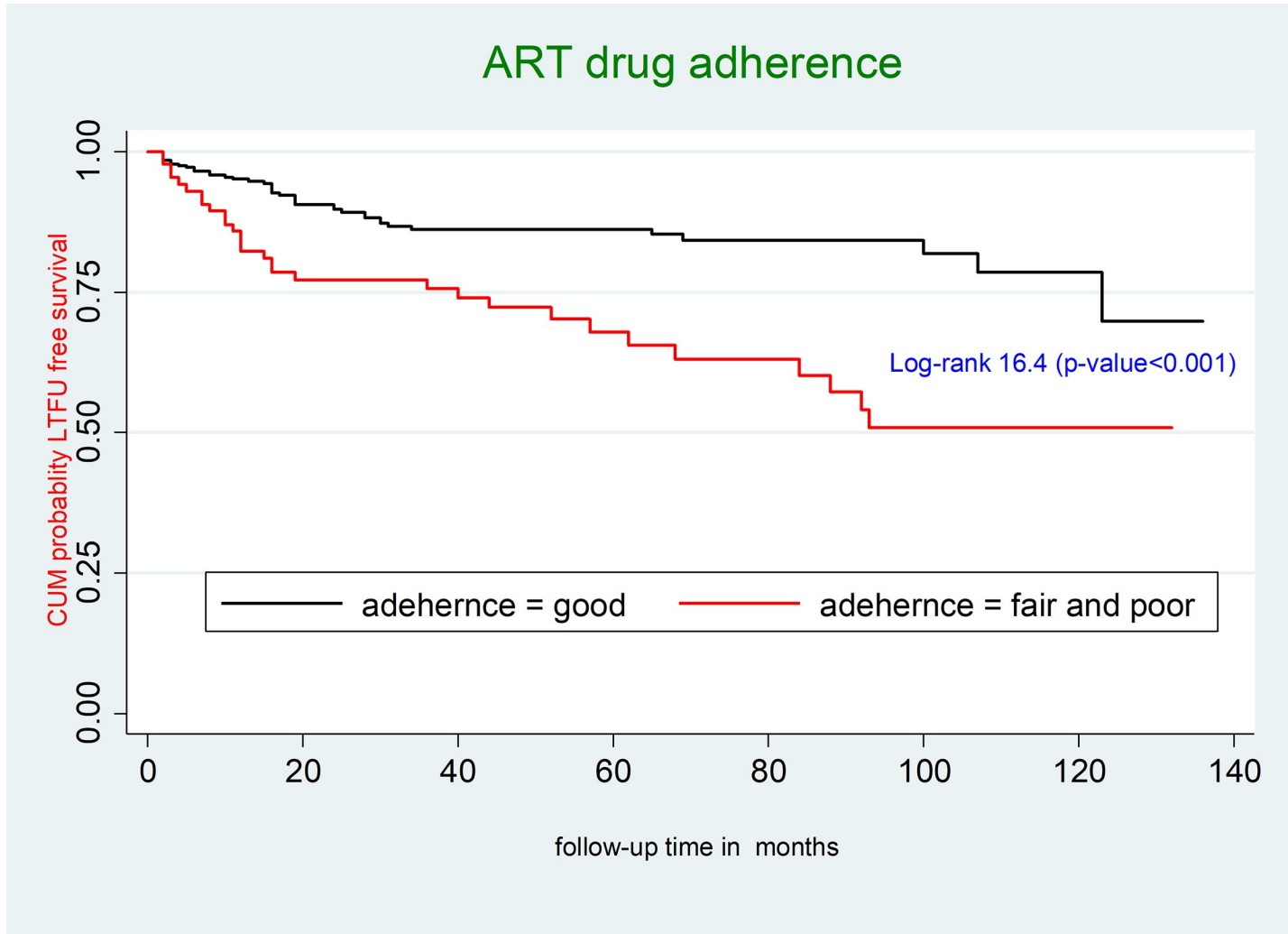

**Fig 3. Kaplan-Meier survival curves to compare LTFU among HIV-infected children receiving ART between good and fair or poor ART drug adherence at Debre-Markos Referral Hospital, Northwest Ethiopia, 2019.**

was 3.2 times (95% CI: 2.0, 5.3) higher than those who were from urban areas. Additionally, the hazard of LTFU among HIV-children who had fair/poor ART drug adherence was 2.3 times (95% CI: 1.4, 3.8) higher as compared to those who had good ART drug adherence prior to three months of last observation. Furthermore, the hazard of LTFU among HIV-infected children started ART with test and treat criteria was 2.7 times (95% CI: 1.4, 5.5) higher than those who started by immunologic and clinical criteria. Finally, the hazard of LTFU among children who took PI-based ART regimens was 2.2 fold (95% CI: 1.1, 4.4) higher as compared to those who took NNRTI based ART regimens (EFV and NVP based) (**Table 3**). The goodness of fit for the cox-proportional hazard model was assessed using a Cox-Snell residual test. The graph suggested that the final model fits the data very well (**Fig 4**).

## Discussion

Nowadays, LTFU among HIV-infected children receiving ART has become a significant public health problem, negatively affecting the treatment outcomes. Therefore, we conducted this

**Table 3. Bi-variable and multivariable Cox regression analysis to identify the predictors of LTFU among HIV-infected children receiving ART at Debre-Markos Referral Hospital, Northwest Ethiopia, 2019.**

| Variables | Survival status | | CHR (95%CI) | AHR (95%CI) |
|---|---|---|---|---|
| | Lost | Censured | | |
| Residence | | | | |
| Urban | 28 | 248 | 1 | 1 |
| Rural | 42 | 90 | 3.6(2.3, 5.9) | 3.2(2.0, 5.3)** |
| Religion of the caregiver | | | | |
| Orthodox | 66 | 283 | 1 | 1 |
| Others | 4 | 55 | 0.31(0.1, 0.9) | 0.5 (0.2, 1.5) |
| Relation of caregiver to the child | | | | |
| Parent | 64 | 284 | 1 | 1 |
| Other guardians | 6 | 54 | 0.5(0.2, 1.1) | 1.1(0.4, 2.8) |
| ART drug adherence | | | | |
| Good | 40 | 281 | 1 | 1 |
| Fair/poor | 30 | 57 | 2.6(1.6, 4.1) | 2.7(1.4, 5.5)** |
| ART eligibility criteria | | | | |
| Immunologic and clinical | 55 | 286 | 1 | 1 |
| Test and treat | 15 | 52 | 2.6(1.4, 4.6) | 2.7(1.4, 5.5) ** |
| Baseline ART regimen | | | | |
| NVP and EFV based | 59 | 311 | 1 | 1 |
| PI based | 11 | 27 | 3.8(1.9, 7.3) | 2.2(1.1, 4.4)** |
| WHO clinical staging | | | | |
| Stage I and II | 31 | 181 | 1 | 1 |
| Stage III and IV | 39 | 157 | 1.3 (0.8, 2.2) | 1.2 (0.7, 2.1) |
| CD4 counts | | | | |
| Below the threshold | 43 | 233 | 1 | 1 |
| Above the threshold | 27 | 103 | 1.5(0.9, 2.4) | 1.5(0.8, 2.6) |
| Wasting | | | | |
| Wasted | 47 | 261 | | |
| Normal | 23 | 77 | 1.5(0.9, 2.5) | 1.3(0.8, 2.2) |
| OI prophylaxis given | | | | |
| Yes | 28 | 200 | 1 | 1 |
| No | 22 | 138 | 0.7(0.4–1.2) | 0.8(0.47, 1.4) |

**significant variables in the multivariable analysis.

retrospective follow-up study to determine the incidence and predictors of LTFU among HIV-infected children receiving ART at Debre Markos Referral Hospital. At the end of follow-up, about 17.1% (95% CI: 13.7, 21.1%) of HIV-infected children lost from ART care. The incidence density of LTFU in this study was estimated to be 4.5 (95% CI: 3.5, 5.7) per 100 child-years of observation. Our finding is in line with studies done in South Africa (5.0 per 100 child-years) [27], Myanmar (4.7 per 100 child-years) [28], and Asia and Africa (4.1 per 100 child-years) [29]. Conversely, our finding is much lower than studies conducted in India (14.4 per 100-child years) [30], South Africa (10.8 per 100 child-years) [31], Tanzania (18.2 per 100-child years) [32], multicountry study (14.2 per 100-child years) [33], Malawi (12.6 per-100 child-years) [19], and Ethiopia (6.22 per 100 child-years) [11].

The above variations could be explained by differences in sample size, study setting, and measurement variability in LTFU. The lower incidence rate of LTFU in this study might be

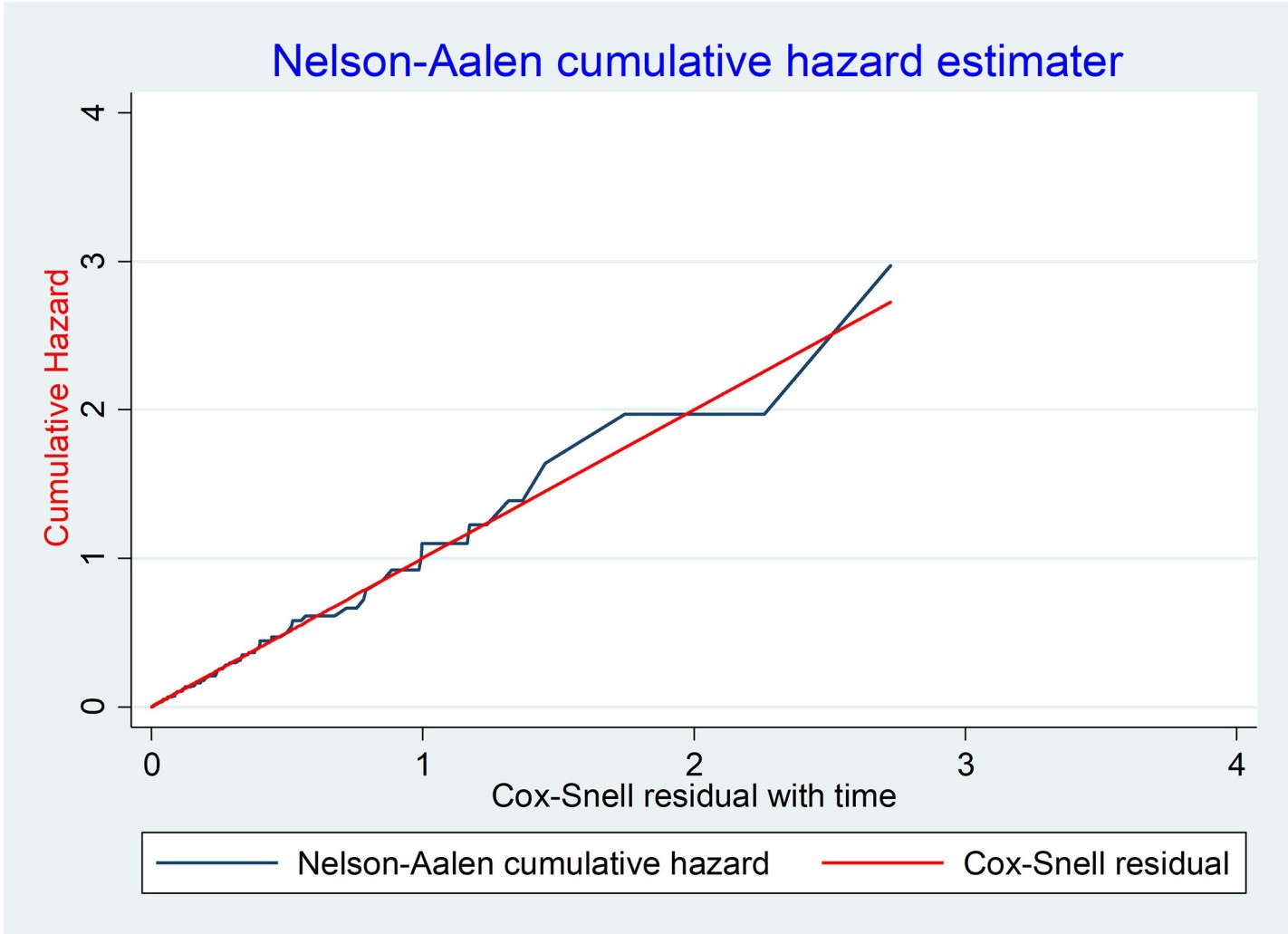

**Fig 4. Assessment of the model fitness using Cox-Snail residual test among HIV-infected children receiving ART at Debre-Markos Referral Hospital, Northwest Ethiopia, 2019.**

due to the different interventions implemented by the Ethiopian government, including adherence support through phone calls, case tracing through a home-to-home visit, and proactive use of community health workers. Another possible justification might be due to the difference in length of follow-up. Our study had relatively longer follow-up time (14 years) compared to previous studies conducted in SSA [19, 28, 31]. A further explanation for the variations could be due to differences in the characteristics of included participants. In this study, more than half (52%) of the study participants were classified as WHO stage I and II; however, about 73.2% of the study participants included in a study from South Africa were classified as WHO stage III and IV [27]. Evidence suggests that HIV-infected children classified as severe disease stage (WHO stage III and IV) at ART initiation were more prone to LTFU as compared to those children classified as WHO stage I and II [11, 31].

This study found that the incidence rate of LTFU within one year of ART initiation was 8.2 (95%, CI: 5.7, 11.7) per-100 child years of observation. However, the incidence of LTFU after one year of ART initiation was 3.3 (95% CI: 2.6, 4.6) per 100 child-years observation. A high rate of LTFU in the early ART phase was observed. This finding is comparable with studies

conducted in South Africa, which showed that LTFU was higher in the first six months of ART follow-up [28, 31]. Supportive findings were also reported from studies done in multicountry based studies [9, 33], Tanzania [32], and Ethiopia [11].

HIV-infected children from rural areas were at higher risk of LTFU as compared to their urban counterparts. This finding is in agreement with other previous studies conducted in Asia and Africa [29], Ethiopia [11], and Malawi [19]. The possible justification could be due to the fact that children living in rural areas travel long distances to get access to ART. Studies suggested that patients living far from a health facility had a greater risk of LTFU [34, 35]. Moreover, children living in rural areas could have difficulties to access favorable transportation; as a result, they could miss their appointments. Additionally, caregivers maybe forget to bring their children to the ART clinic at each visit. Lastly, the caregiver's level of education has a significant effect on LTFU because most rural mothers are uneducated. As a result, they commonly seek faith healing or traditional therapies [36].

This study also indicated that children who had fair or poor ART drug adherence were at higher risk of LTFU compared to those who had good adherence. It is well understood that the relationship between ART drug adherence and LTFU is bidirectional. The common reasons for poor ART adherence among people living with HIV are the use of traditional/herbal medicine, dissatisfaction with healthcare services, depression, discrimination and stigmatization, and poor social support [37]. These factors are also directly related to the risk of LTFU. The other common reason for LTFU is fear of ART side-effects. Lastly, if patens do not take their drug properly (have poor ART adherence), they don't have good improvements in the early ART phase. This leads to early treatment failure and the rapid development of drug resistance.

This study also found that children who started ART based on the test and treat approach were at higher risk of LTFU as compared to those who started ART based on immunologic (CD4 counts) or clinical (WHO staging) criteria. Despite WHO recommends that HIV-positive patients can start ART within the seven days after HIV-confirmation, evidence from an observational study revealed that starting ART on the same day of HIV diagnosis increases the risk of LTFU [38]. According to the WHO recommendation, higher LTFU is documented as the main weakness of the test and treat approach [39]. Another possible explanation could be due since starting ART at the same day of HIV confirmation without intensive counselling and adequate preparation may lead to fear of stigma and discrimination and finally leads to LTFU in the early ART phase.

Finally, children who started PI-based ART regimens were at higher risk of LTFU as compared to those who started non-nucleoside reverse transcriptase inhibitors (Nevirapine or Efavirenz) based ART regimens. This finding contradicts a previous study done in Myanmar [28]. This variation could be due to the baseline differences between study participants at ART initiation, leading to the prescription of PI-based regimens. The actual effects of PI-based regimens on better treatment outcomes would need further follow-up studies.

## Potential limitations of the study

This study has some constraints that must be considered before interpreting results. As the study used secondary data, some important variables such as viral load, micronutrient deficiency, and immunization status of the child were not included. This study was also unable to ascertain the reasons for and outcomes of LTFU due to incomplete documentation. Patients recorded as LTFU might be died or started ART in another health institutions. Therefore, the actual LTFU might be overestimated in this study. The study's main strength was conducted for a more extended period of follow-up time (14 years); this could increase observation time.

## Conclusion

This study found that one in every six HIV-infected children lost form ART follow-up. Relatively a lower rate of LTFU was observed as compared to previous studies reported in Ethiopian and other SSA countries. Moreover, a higher rate of LTFU was seen within the first year of ART initiation. HIV-infected children from rural areas, having fair or poor ART drug adherence, started ART based on test and treat approach, and taking PI-based ART regimens were at higher risk of LTFU. Thus, particular emphasis and close follow-up must be given within the first year of ART initiation. Moreover, tracing mechanisms should be strengthened for children who are from rural areas. Furthermore, providing participatory advice and letting them decide rather than enforcing to start ART immediately after HIV confirmation is highly appreciated. Further prospective follow-up studies by considering viral load, child immunization, and micronutrient deficiencies are highly recommended. Lastly, qualitative studies to explore the reasons of LTFU are also recommended.

## Supporting information

**S1 File. This S1 File is the date set used for this study.**
(DTA)

## Acknowledgments

We would like to acknowledge the healthcare professionals working in the ART clinic of Debre Markos Referral Hospital for their generous support during data collection and chart retrieval. We also extend our heartfelt thanks to data collectors.

## Author Contributions

**Conceptualization:** Yitbarek Tenaw Hibstie.

**Data curation:** Yitbarek Tenaw Hibstie, Belisty Temesgen.

**Formal analysis:** Yitbarek Tenaw Hibstie, Belisty Temesgen.

**Investigation:** Mamaru Wubale Melkamu.

**Methodology:** Yitbarek Tenaw Hibstie, Animut Alebel.

**Project administration:** Yitbarek Tenaw Hibstie, Animut Alebel.

**Software:** Yitbarek Tenaw Hibstie, Getiye Dejenu Kibret.

**Supervision:** Getiye Dejenu Kibret, Asmare Talie, Belisty Temesgen, Animut Alebel.

**Validation:** Asmare Talie, Mamaru Wubale Melkamu, Animut Alebel.

**Visualization:** Getiye Dejenu Kibret, Asmare Talie.

**Writing – original draft:** Yitbarek Tenaw Hibstie.

**Writing – review & editing:** Getiye Dejenu Kibret, Belisty Temesgen, Mamaru Wubale Melkamu, Animut Alebel.

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
