## [Decision Letter · Decision Letter 0]

13 Aug 2020

PONE-D-20-16100

Nearly one in every six children lost from ART follow-up at Debre Markos Referral Hospital, Northwest Ethiopia: A 14-year retrospective follow-up study

PLOS ONE

Dear Dr. Animut Alebel

Thank you for submitting your manuscript to PLOS ONE. After careful consideration, we feel that it has merit but does not fully meet PLOS ONE’s publication criteria as it currently stands. Therefore, we invite you to submit a revised version of the manuscript that addresses the points raised during the review process.

We look forward to receiving your revised manuscript.

Kind regards,

Claudia Marotta

Academic Editor

PLOS ONE

Journal Requirements:

2. We noticed you have some minor occurrence of overlapping text with the following previous publication, which needs to be addressed:

https://journals.plos.org/plosone/article?id=10.1371%2Fjournal.pone.0195435

In your revision ensure you cite all your sources (including your own works), and quote or rephrase any duplicated text outside the methods section. Further consideration is dependent on these concerns being addressed.

Additional Editor Comments (if provided):

Dear Authors,

I appreciate a lot your manuscript.

Following reviewes suggestions the paper can be publish after minor revisions

Reviewers' comments:

Reviewer's Responses to Questions

**Comments to the Author**

1. Is the manuscript technically sound, and do the data support the conclusions?

Reviewer #1: Yes

Reviewer #2: Yes

2. Has the statistical analysis been performed appropriately and rigorously? 

Reviewer #1: Yes

Reviewer #2: Yes

3. Have the authors made all data underlying the findings in their manuscript fully available?

Reviewer #1: Yes

Reviewer #2: Yes

4. Is the manuscript presented in an intelligible fashion and written in standard English?

Reviewer #1: Yes

Reviewer #2: Yes

5. Review Comments to the Author

Reviewer #1: Authors written a very interesting article on a vunerable population (childeren) in a fragile setting .

Only some suggestion:

1. Introduction: well wrote, introduce the concept of children at risk (cite this article. The At Risk Child Clinic (ARCC): 3 Years of Health Activities in Support of the Most Vulnerable Children in Beira, Mozambique. Int J Environ Res Public Health. 2018;15(7):1350. Published 2018) that explain the role also of the research on this issue. Als see this article e Schacht C, Lucas C, Mboa C, et al. Access to HIV prevention and care for HIV-exposed and HIV-infected children: a qualitative study in rural and urban Mozambique. BMC Public Health. 2014;14:1240. Published 2014 Dec 3.

2. Methods and Results: no comment.

3. Discussion: well. I f you can improve with a proposal. Why the children lost to follow up? discuss other experience as for example this in Mozambique (Pathways of care for HIV infected children in Beira, Mozambique: pre-post intervention study to assess impact of task shifting. BMC Public Health. 2018;18(1):703. Published 2018 Jun 7.)

I consider it as an excellent article.

Reviewer #2: This institution-based retrospective follow-up study conducted among 408 HIV-infected children on ART at Debre Markos Referral Hospital between 2005 and March 15, 2019, has provided very important information or indicators for lost to follow up among kids enrolled in HIV treatment programs that could be generalized or applied to other settings within Ethiopia and countries with similar status. Lost to follow up is current a serious problem in the management of HIV treatment among adults and kids, putting together data to identify potential reasons for this in this setting is highly rewarding as it will definitely lead t implementation of strategies to address gaps identified. It is interesting to identify in the study that starting ART immediately or sooner after diagnosis following WHO recommendation is also a factor that influence lost to follow. Addressing should also impact national policy on the country’s treatment program. This study is highly recommended. Potential limitations, including not being able to include that of VL which could provided information to virally suppression status of these kids is nothing; something that should be considered in future studies.

6. PLOS authors have the option to publish the peer review history of their article (what does this mean?). If published, this will include your full peer review and any attached files.

Reviewer #1: **Yes: **Francesco Di Gennaro

Reviewer #2: No

---

## [Author Response · Author response to Decision Letter 0]

25 Aug 2020

1) Thank you for updating your data availability statement. You note that your data are available within the Supporting Information files, but no such files have been included with your submission. At this time we ask that you please upload your minimal data set as a Supporting Information file, or to a public repository such as Fig share or Dryad. Please also ensure that when you upload your file you include separate captions for your supplementary files at the end of your manuscript. As soon as you confirm the location of the data underlying your findings, we will be able to proceed with the review of your submission.

Response: Thank you for your concern. We have uploaded the Stata data set as S1 file.

2) Your ethics statement must appear in the Methods section of your manuscript. If your ethics statement is written in any section besides the Methods, please move it to the Methods section and delete it from any other section. Please also ensure that your ethics statement is included in your manuscript, as the ethics section of your online submission will not be published alongside your manuscript.

Response: Thank you for your concern. We have moved the ethics statement to the method section. Please see line 184-193. 

Reviewer #1: 

Authors written a very interesting article on a vulnerable population (children) in a fragile setting.

Response: Thank you for your encouraging words.

Only some suggestion: 

1. Introduction: well written, introduce the concept of children at risk (cite this article. The At Risk Child Clinic (ARCC): 3 Years of Health Activities in Support of the Most Vulnerable Children in Beira, Mozambique. Int J Environ Res Public Health. 2018;15(7):1350. Published 2018) that explain the role also of the research on this issue. Also see this article e Schacht C, Lucas C, Mboa C, et al. Access to HIV prevention and care for HIV-exposed and HIV-infected children: a qualitative study in rural and urban Mozambique. BMC Public Health. 2014;14:1240. Published 2014 Dec 3.

Response: Thank you for your suggestion. We have introduced the concept of “child at risk” in the first paragraph of our introduction. Please see line 52-54.

2. Methods and Results: no comment.

 Response: Thank you for your precious time.

3. Discussion: well. I f you can improve with a proposal. Why the children lost to follow up? Discuss other experience as for example this in Mozambique (Pathways of care for HIV infected children in Beira, Mozambique: pre-post intervention study to assess impact of task shifting. BMC Public Health. 2018;18 (1):703. Published 2018 Jun 

Response: Thank you very much for your suggestion. The suggested article is not related to our title. Therefore, we did not cite it. The question “Why the children lost to follow up?” needs further studies. As this is beyond our objective, we haven’t any data on this issue. This has been acknowledged as a limitation part of our research. And further qualitative studies are recommended to address this issue. Please see line 335-36 and 351-52.

 I consider it as an excellent article.

Reviewer #2: 

This institution-based retrospective follow-up study conducted among 408 HIV-infected children on ART at Debre Markos Referral Hospital between 2005 and March 15, 2019, has provided very important information or indicators for lost to follow up among kids enrolled in HIV treatment programs that could be generalized or applied to other settings within Ethiopia and countries with similar status. Lost to follow up is current a serious problem in the management of HIV treatment among adults and kids, putting together data to identify potential reasons for this in this setting is highly rewarding as it will definitely lead to implementation of strategies to address gaps identified. It is interesting to identify in the study that starting ART immediately or sooner after diagnosis following WHO recommendation is also a factor that influence lost to follow. Addressing should also impact national policy on the country’s treatment program. This study is highly recommended. Potential limitations, including not being able to include that of VL which could provide information to virally suppression status of these kids is nothing; something that should be considered in future studies.

Response: Thank you for your concern. Future studies by considering VL are strongly recommended. Please see line 349-351.

---

## [Decision Letter · Decision Letter 1]

28 Aug 2020

Nearly one in every six children lost from ART follow-up at Debre Markos Referral Hospital, Northwest Ethiopia: A 14-year retrospective follow-up study

PONE-D-20-16100R1

Dear Dr. Alebel,

We’re pleased to inform you that your manuscript has been judged scientifically suitable for publication and will be formally accepted for publication once it meets all outstanding technical requirements.

Kind regards,

Claudia Marotta

Academic Editor

PLOS ONE

Additional Editor Comments (optional):

Dear Authors,

two different reviewers suggest to accept your manuscript.

I appreciate a lot element of your article: the long period-time of study (14ys), the setting (Ethiopia) and the idea research (child, HIV, lost to follw up)

Congratulations for your great job!

Reviewers' comments:

Reviewer's Responses to Questions

**Comments to the Author**

1. If the authors have adequately addressed your comments raised in a previous round of review and you feel that this manuscript is now acceptable for publication, you may indicate that here to bypass the “Comments to the Author” section, enter your conflict of interest statement in the “Confidential to Editor” section, and submit your "Accept" recommendation.

Reviewer #1: All comments have been addressed

2. Is the manuscript technically sound, and do the data support the conclusions?

Reviewer #1: Yes

3. Has the statistical analysis been performed appropriately and rigorously? 

Reviewer #1: Yes

4. Have the authors made all data underlying the findings in their manuscript fully available?

Reviewer #1: Yes

5. Is the manuscript presented in an intelligible fashion and written in standard English?

Reviewer #1: Yes

6. Review Comments to the Author

Reviewer #1: Congratulations I appreciate a lot your article. I appreciate the methods and the idea reasearch.

7. PLOS authors have the option to publish the peer review history of their article (what does this mean?). If published, this will include your full peer review and any attached files.

Reviewer #1: **Yes: **Di Gennaro Framcesco

---

## [Editor Report · Acceptance letter]

3 Sep 2020

PONE-D-20-16100R1 

Nearly one in every six HIV-infected children lost from ART follow-up at Debre Markos Referral Hospital, Northwest Ethiopia: A 14-year retrospective follow-up study  

Dear Dr. Alebel:

I'm pleased to inform you that your manuscript has been deemed suitable for publication in PLOS ONE. Congratulations! Your manuscript is now with our production department. 

Kind regards, 

on behalf of

Dr. Claudia Marotta 

Academic Editor

PLOS ONE